# iCardioMonitor Digital Monitoring System for People with Heart Failure: Development and Evaluation of Its Accessibility and Usability

**DOI:** 10.3390/healthcare12191986

**Published:** 2024-10-05

**Authors:** Set Perez-Gonzalez, Maria del Mar Fernandez-Alvarez, Noemi Gutierrez-Iglesias, Beatriz Díaz-Molina, Vanesa Alonso-Fernandez, Ruben Martin-Payo

**Affiliations:** 1Department of Mathematics, University of Oviedo, 33003 Oviedo, Spain; perezset@uniovi.es; 2Precam Research Group, Faculty of Medicine and Health Sciences, Instituto de Investigación Sanitaria del Principado de Asturias, University of Oviedo, 33003 Oviedo, Spain; martinruben@uniovi.es; 3Department of Cardiology, Hospital Universitario Central de Asturias, 33011 Oviedo, Spain; uo176140@uniovi.es (N.G.-I.); beadimo@secardiologia.es (B.D.-M.); vanesa.alonso@sespa.es (V.A.-F.)

**Keywords:** eHealth, heart failure, knowledge-base algorithm

## Abstract

**Background/Objectives:** The use of eHealth as a monitoring system in people with heart failure (HF) has been shown to be effective in promoting self-care and reducing re-admissions and mortality. The present study develops and evaluates the accessibility and usability of the web app iCardioMonitor HF monitoring system. **Methods:** This study consisted of two stages. The first stage (co-design) comprised two phases: (1) analysis of the scientific literature and expert opinions and (2) co-design of the iCardioMonitor (web app plus a knowledge-base algorithm) and definition of alert criteria. The second stage (cross-sectional descriptive study) analyzed system accessibility (% of people using the iCardioMonitor and % of parameters recorded) and usability, employing the Spanish version of the System Usability Scale for the Assessment of Electronic Tools. **Results:** The iCardioMonitor was configured by a web app and an algorithm with the capacity to detect decompensated HF automatically. A total of 45 patients with an average age of 55.8 years (standard deviation [SD] = 10.582) and an average time since diagnosis of 7.1 years (SD = 7.471) participated in the second stage. The percentage of iCardioMonitor use was 83.2%. The average usability score was 77.2 points (SD = 21.828), higher in women than men (89.2; SD = 1.443–76.0; SD = 1.443) (*p* = 0.004). The usability score was higher the shorter the time since diagnosis (r = 0.402; *p* = 0.025) and the higher the number of responses (r = 0.377; *p* = 0.031). **Conclusions:** The results obtained show that iCardioMonitor is a tool accepted by patients and has obtained a remarkable score on the usability scale. iCardioMonitor was configured by a web app and an algorithm with the capacity to detect decompensated HF automatically.

## 1. Introduction

The increase in life expectancy, progress in healthcare, and the rise in unhealthy lifestyles such as sedentarism, smoking, and inadequate eating habits have caused chronic diseases to be the predominant epidemiological model in Spain. Of these diseases, cardiovascular disorders are the leading cause of death [1,2]. The World Health Organization (WHO, Geneva, Switzerland) [3] estimated that in 2019, 17.9 million people died from this cause, representing approximately one-third of all deaths worldwide.

One of the most prominent cardiovascular diseases is heart failure (HF). It is estimated that over 64 million people worldwide and 6 million people in Europe have been diagnosed with HF [4,5]. In Spain, recent data indicate that the incidence rate of HF is 2.78 cases/1000 persons/year [6], constituting the main cause of hospital admissions among people over 65 years of age and representing 2–3% of global health costs [7].

Decompensated HF is a frequent cause of patient admission in hospitals and is generally preceded by an exacerbation of symptoms. In order to reduce hospital re-admissions and the mortality rate among this group, it is essential to implement self-care programs or practices to maintain health through preventive and health-promoting conducts [8,9]. HF self-care requires patients to perform daily self-monitoring of symptoms, signs, and changes in body weight, as well as to adhere to the prescribed medication, diet, physical activity, and follow-ups [8,10]. 

Telemedicine has become a valuable resource in the daily management of HF patients. The integration and development of diverse systems, from basic monitoring devices to portable technology and state-of-the-art remote monitoring systems, play an important role in the care of these patients [11]. 

Currently, some home-based eHealth interventions designed for these patients already incorporate monitoring systems. Telemonitoring allows patients to remotely provide digital health information to support or optimize their care. There are several telemonitoring systems currently available for HF patients. Home telemonitoring can help maintain quality of care when needed as well as reduce costs and the number of patient visits to the healthcare facility. Other systems are designed to provide support at specific times when the patient requests help or care. There are also implanted therapeutic devices that can immediately and remotely provide information about the device or physiological aspects of the patient. Implantable loop recorders can be injected subcutaneously and used to monitor heart rate and rhythm, activity, and bioimpedance [12].

Systems incorporating monitoring have been shown to be effective in increasing patient knowledge [13], motivating self-care [14], and obtaining better health outcomes, with a decrease in all-cause mortality, hospital admissions or re-admissions, and improved quality of life [8,15,16]. 

Heterogeneity in program designs and content can be observed in the literature. Particularly noteworthy are strategies based on e-Health, as they remove some of the barriers that have traditionally limited self-care programs [8] and take into account established patient preference for proactive systems that gather information in real-time [17,18,19]. In terms of content, there seems to be a consensus that they should provide health-promoting educational information and health data collection and monitoring [20]. Last, Li et al. [20] suggest that these strategies should focus on the specific needs of the population for which they are designed.

It thus seems that systems designed to be used by people with HF, which include education focusing on disease management, self-care, and symptom monitoring, are effective and beneficial for patients [13]. However, their heterogeneity in design and content suggests that they should be developed based on the needs of the population; hence the importance of co-design, i.e., considering and encouraging the participation of end-users of such systems in their development and evaluating the usability of the systems before assessing their effectiveness [20,21]. 

With this in mind, the present study was carried out to develop and evaluate the accessibility and usability of a novel digital system iCardioMonitor (web app plus an algorithm), in Spanish, for patients with HF, with the aim of improving their knowledge of the signs and symptoms of HF, and to facilitate their self-care and monitoring.

## 2. Materials and Methods

### 2.1. Design

This study was divided into two stages. The first consisted of the co-design of the iCardioMonitor system (web app plus a knowledge-base algorithm), and the second comprised the analysis of its accessibility and usability. This second stage involved a cross-sectional descriptive study design. 

### 2.2. Study Population

For the development of the co-design stage, the collaboration of cardiology experts, physicians, or nurses, recruited on an intentional basis, was sought, considering the following inclusion criteria: (i) at least 5 years of experience in HF and (ii) currently working in an HF unit. 

A sample population was also selected to take into account their experiences and perspectives in the development of the system. Although the sample was selected on an opportunistic basis, the following criteria were considered in order to obtain a sample with characteristics similar to those of end users: (i) aged over 65 and (ii) different educational levels. People with physical or cognitive limitations that would prevent them from participating in this study were excluded, as were those with insufficient Spanish fluency.

For the accessibility and usability analysis, the study population consisted of patients diagnosed with HF who were being followed at the Heart Failure Unit of the Hospital Universitario Central de Asturias (Spain). The following inclusion criteria were considered: (i) a complex HF diagnosis and (ii) capacity to access the web app. The exclusion criteria were: (i) patients lacking devices to access the web app; (ii) insufficient Spanish fluency; (iii) patients who were admitted to the hospital or who changed the follow-up center during the study. 

Usability is defined as the *“extent to which a system, product or service can be used by specified users to achieve specified goals with effectiveness in a specified context of use”* [22] or, in other words, without errors occurring during its use. Considering the usability of the interface as the main variable, and according to Cazañas et al. [23], a minimum sample of 25 patients would be needed to reliably detect 99% of the potential errors (e.g., wrong data recording). 

Recruitment took place in cardiology hospital admission units and consulting rooms, coinciding with patient follow-up or a hospital stay. The HF team was in charge of patient recruitment. The purpose of this study was explained to the patients during recruitment. 

Those patients who agreed to participate signed the corresponding informed consent form (ICF). After signing the ICF, the patients received two questionnaires: (1) a questionnaire for personal data collection (age, gender, educational level, marital status, family support, cohabitation, time since HF diagnosis) and (2) a second questionnaire on self-care based on the European Heart Failure Self-care Behaviour Scale (EHFScB) [24]. After completing these questionnaires, the patients received instructions on how to use the web app and any doubts were resolved. Subsequently, access to the web app was provided for a period of 30 days. Each patient also received an automatic blood pressure monitor for daily recordings, along with an e-mail address for technical help, if needed, or error reporting. 

After 30 days of use, the patients were contacted by telephone to indicate the end of the period of use of the web app. On the other hand, two questionnaires were sent again by letter or e-mail (according to patient choice) to quantify self-care based on the European Heart Failure Self-care Behaviour Scale (EHFScB) [24] and the usability of the web app using the Spanish Version of the System Usability Scale (SUS) for the Assessment of Electronic Tools [25].

### 2.3. Development of the iCardioMonitor System (Stage 1)

The iCardioMonitor consisted of a web app and a knowledge-based algorithm. The web app included the following: (1) Information on behaviors to be adopted by patients with HF; (2) adequate measurement of blood pressure; (3) a self-report registry of signs and symptoms compatible with decompensated HF to be completed by the patients on a daily basis. Data recorded by the patients were automatically analysed by an algorithm developed ad hoc for the iCardioMonitor and capable of detecting decompensated HF symptoms (alerts). Any detected alert was automatically transmitted to the Department of Cardiology, which in turn contacted the patient. 

The objective of this stage (development of the iCardioMonitor system) was to determine what information should be included in the web app, what items should be included in the form, and under what conditions an alert should be sent to the Department of Cardiology (i.e., alert definition).

The development of the iCardioMonitor digital system was guided by the principles of the Unified Theory of Acceptance and Use of Technology [26], which predicts patient willingness to use technology. This specifically refers to performance expectancy, effort expectancy, social influence, and facilitating conditions.

Co-design of the iCardioMonitor comprised two phases: (1) analysis of the scientific literature and expert opinions and (2) design of the iCardioMonitor web app and definition of alert criteria.

Patients were pseudo-anonymized to ensure confidentiality. The patients employed a username and password to access the web. Only cardiology staff could link the user code to the patient in question if necessary (e.g., on receipt of an alert from the iCardioMonitor). 

#### 2.3.1. Phase 1: Analysis of the Scientific Literature and Expert Opinions

The selection of content to be included in the iCardioMonitor web app was based on a literature review [6,9,12,27,28]. Specifically monitoring indicators such as weight, blood pressure, symptoms, and heart rate [6,9,12,27,28] and lifestyle recommendations [27]. In addition, recommendations from experts (3 cardiologists and a cardiology area nurse, all with over 5 years of professional experience in HF). The process for the inclusion of information followed the following steps: (i) literature review and expert input in order to identify content to be included in the web app, both monitoring and information; (ii) a check-list was prepared, where all the items identified in the previous step were included. Each item included a response option on a Likert-type scale (not recommendable to highly recommendable); (iii) the checklist was provided to the experts so that they could evaluate the validity of the items, eliminating those evaluated as “not recommendable.” Based on the above, the decision was made to include the following content: (i) information on the signs and symptoms of decompensated HF, (ii) lifestyle recommendations for pre-venting decompensation, and (iii) how to measure blood pressure adequately. 

#### 2.3.2. Phase 2: Design of the iCardioMonitor Web App and Definition of Alert Criteria 

The draft of the web app includes three sections. The first section includes information based on the literature and expert recommendations obtained in Phase 1 related to a healthy lifestyle and signs and symptoms of decompensated HF. The objective of this section was to make it possible for the patients to answer the following questions: What is heart failure? What are the main symptoms? How should I take care of myself? The second section focuses on instructing the patients to measure their blood pressure correctly.

The third section included the patient self-report registry of signs and symptoms. The registry form was made for patients to record information related to the signs and symptoms of decompensated HF on a daily basis, thus contributing to their daily self-care, HF monitoring, and real-time detection of potential decompensation episodes. The self-report registry included the following items: current body weight (kg), medication use the day before (yes/no), need to sleep with a cushion or elevation of the headrest the night before (yes/no), breathing difficulties (yes/no), systolic and diastolic blood pressure (mmHg), heart rate, presence of fatigue (yes/no), swollen feet (yes/no) [20,29]. In sum, these are the input variables that patients must enter into the web app and are, therefore, the ones considered by the algorithm. 

Once the draft of the web app had been developed, 7 individuals with the same characteristics as the end users were requested to evaluate the functionality, and the same experts who participated in Phase 1 reviewed the content.

### 2.4. Evaluation of Accessibility and Usability (Stage 2)

Recruitment took place in cardiology hospital admission units and consulting rooms, coinciding with patient follow-up or a hospital stay. The patients received an explanation about the purpose of this study and were invited to participate.

Those patients who gave verbal consent to participation, in turn, signed the corresponding informed consent form (ICF).

After signing the ICF, the patients were scheduled for a visit by the HF team. During that visit, the patients completed a questionnaire for personal data collection (age, gender, educational level, marital status, family support, cohabitation, and time since HF diagnosis) and the European Heart Failure Self-care Behaviour Scale (EHFScB) [24]. The scale consisted of a self-administered 12-item questionnaire addressing different aspects of patient self-care. Each item was rated from 1 (strongly/always agree) to 5 (strong/always disagree). The total score, therefore, could range from 12 (best self-care) to 60 (worst self-care). 

After completing the above, the patients received instructions on how to use the web app and were given access to it for 30 days. 

Each patient also received an automatic blood pressure monitor for daily recordings, along with an e-mail address for technical help, if needed, or error reporting. 

After 30 days of use, the accessibility and usability of the web app were evaluated. 

The accessibility of the system was evaluated using the following indicators: % of people who used the iCardioMonitor; % registry of HF monitoring parameters in the iCardioMonitor digital system (patients recording data/day)—the range being 0–1350 registries. This information was obtained directly from the records that were automatically produced after patients’ use of iCardioMonitor.

The usability of the iCardioMonitor was evaluated as perceived by the users and measured with the Spanish version of the Usability System Scale for the Assessment of Electronic Tools (SUS) [25] (Cronbach α = 0.812) [27]. The questionnaire consisted of 10 items scored on a 5-point Likert scale from 1 (Strongly disagree) to 5 (Strongly agree). The questionnaire was based on positive and negative affirmations. The odd item score was obtained by subtracting 1 from the item score as given by the user, while the even item score was obtained by subtracting the item score given by the user from 5. The total score was calculated by adding up the odd and even scores, multiplied by 2.5. The final score ranged from 0 (worst usability) to 100 (best usability), where excellent usability corresponded to a score of >85 and good usability to a score of 68–84 [25].

In addition, the situations in which the algorithm identified an alert and the correlation between the definition of an alert and what was identified as an alert were analyzed.

### 2.5. Ethical Points

This study complied with the Declaration of Helsinki and was approved by the Clinical Research Ethics Committee of Hospital Universitario Central de Asturias (2023.007). All the participants signed the ICF to be included in this study.

### 2.6. Data Analysis

An anonymized database was created, where a numerical code was assigned to each participant. Analyses were performed using the SPSS IBM statistical package, version 27.0. Only values of *p* < 0.05 were considered statistically significant.

A descriptive analysis was performed on the personal parameters, acceptability, and usability, reporting the average and standard deviation (SD) for quantitative variables and percentages for qualitative variables. 

A normal data distribution was confirmed by the Shapiro–Wilk test. 

Pearson correlation tests were performed between usability scores and age, time since diagnosis, self-care and number of responses. The unpaired t-test (dichotomous variables) or analysis of variance (ANOVA) (more than two groups) was used to compare the average usability scores based on patient characteristics.

Last, a linear regression analysis was performed with the usability score as the dependent variable and the personal parameters, self-care score, and number of responses as independent variables.

## 3. Results

### 3.1. Stage 1: Development of the iCardioMonitor System

After the analysis of the scientific literature and expert opinions, the iCardioMonitor finally consisted of a web app including three sections and an algorithm with the capacity to automatically detect and report signs of decompensated HF to the Department of Cardiology.

The first section of the web app includes information on healthy lifestyle habits and signs and symptoms of decompensated HF. The second section focuses on instructing the patients to measure their blood pressure correctly. The third section includes the patient self-report registry of signs and symptoms. 

The algorithm used to detect warning signs included the following warning criteria: weight gain ≥ 2%/week; use of medication the day before = no; need to sleep with a pillow or elevated headrest the night before = yes; dyspnoea = yes; systolic blood pressure ≥ 180 and diastolic blood pressure ≥ 100; presence of fatigue = yes; swollen feet = yes.

Once the first version of the web app had been developed, 7 individuals with the same characteristics as the end users—mean age 66 years (standard deviation [SD] 7.51), 28.6% with primary education, 57.1% with secondary education and 14.3% with university education—were requested to evaluate the functionality of the form. For this purpose, they received access to the form and were asked to enter fictitious data to check correct compliance and, thus, the reliability of the data registry based on the percentage coincidence between provided and recorded data. The resulting percentage was 98%.

Last, the same experts who participated in Phase 1 reviewed the content, gave their approval (100% agreement), and defined the alert criteria to be included in the algorithm.

### 3.2. Stage 2: Evaluation of Accessibility and Usability

#### 3.2.1. Description of the Study Population

The sample for evaluating accessibility and usability consisted of 45 patients. The average age was 55.8 years (standard deviation [SD] = 10.582), and the average time since diagnosis was 7.1 years (SD = 7.471). Most of the participants were males (88.9%), with secondary education (50.0%), married or with a partner (63.6%), cohabiting (86.4%), and retired or unemployed (81.8%). The average score on the EHFScB self-care scale was 30.0 points (SD = 6.654) (Table 1).

#### 3.2.2. Acceptability

There were 1080 registries, of which 1008 were valid and 72 had errors (6.7%). Of the 45 patients who started the study, 12 withdrew before the first month (26.6%). In 50% of these cases, the reason for withdrawal was hospital admission or health issues. Of the twelve patients, eight had used the tool for 1–10 days, two for 11–20 days, and two for 25 days. 

Of the 33 patients that completed the 30-day period, 13 (39.9%) used the tool for 1–10 days, 3 (9.1%) for 11–20 days, and 17 (51.5%) used it more than 21 times. Of the latter, 12 patients recorded all 30 measurements. 

Considering the 45 participants, the percentage of global use from the start to withdrawal (*n* = 12) or the end (*n* = 33) was 83.2%.

#### 3.2.3. Usability

The average usability score was 77.2 points (SD = 21.828); according to SUS [25], this means good usability. There were no significant differences in terms of age (*p* = 0.085), self-care capacity (*p* = 0.227), educational level (*p* = 0.108), or cohabitation (*p* = 0.385). Statistical significance was observed in relation to gender (*p* = 0.004), with women (mean = 89.2; SD = 1.443) having a higher perception of usability than men (mean = 76.0; SD = 1.443). Similarly, in terms of time since diagnosis, the perception of usability was significantly greater at shorter times since diagnosis (*p* = 0.025), and significance was also observed in terms of the number of responses (*p* = 0.031).

## 4. Discussion

The accessibility indicator results obtained show that the iCardioMonitor can be used by the population for monitoring the symptoms indicating worsening of HF. Likewise, the usability scores suggest high usability independently of the personal characteristics of the users. A shorter time since HF diagnosis and an older patient age were identified as usability predictors. 

Development of the iCardioMonitor web app involved a rigorous process following the descriptions of Guilabert et al. [30] and Mandracchia et al. [31] and considering the applicable essential elements defined by Llorens-Vernet et al. [32], such as for example security, content and usability. Special consideration was given to the content of the web app, healthy lifestyle habits, and signs and symptoms of decompensated HF. This appears logical in the case of the iCardioMonitor since one of its purposes is to transmit knowledge to allow the population to take greater control of the disease—this is a factor identified in the literature as being crucial in promoting self-care [8,33]. It should be emphasized that using scientific evidence guarantees the quality of the content and is one of the aspects regarded as most important by both patients and healthcare professionals [31]. In our opinion, this is one of the positive aspects to be noted in the case of the iCardioMonitor. The content of the iCardioMonitor was judged by experts as highly appropriate and was based on the systematic analysis of the existing scientific literature [6,9,12,20,27,28]. 

On the other hand, the review carried out by Jakob et al. [34] found the adaptation of content to the patient’s needs was one of the factors influencing adherence to the use of digital tools designed for the management of non-transmissible diseases. As mentioned above, the content of the iCardioMonitor was established in consultation with professionals and patients, thereby adapting it to the demands of both. This may have been decisive in generating the observed accessibility results. 

It is important to highlight the need to promote the acquisition of knowledge, for example, about lifestyle measures for preventing HF complications or symptoms that predict worsening of HF. In fact, knowledge is one of the factors that promote self-care behaviors in HF [35,36], and this can contribute to reducing hospital admissions and mortality associated with HF [36].

Another important aspect of our study was the high usability scores obtained, which were very satisfactory (77.2%). The evidence suggests that there is a significant association between adequate usability and easier tool usage [37,38], adherence to the use of the web app [32], effectiveness and efficiency in achieving the expected goals [39], and increased patient safety [40,41]. In a study carried out by Bylappa et al. [42], describing the usability and feasibility results of an app similar to iCardioMonitor, usability was seen to be slightly greater, though it should be noted that the participants reported difficulty in completing some items, in contrast to the situation in our own study.

On the other hand, mention should be made of a study published by Floegel et al. [43]. In this case, monitoring, as suggested by the authors, included an ankle monitor with automatic data recording. Although percentage participation during the study, and thus the data registry, was slightly greater (92%) than in our case (83.2%), after completing the study, only 18% of the subjects expressed a willingness to continue using the ankle monitor.

The recorded scores may be related to the fact that the principles of the Unified Theory of Acceptance and Use of Technology were taken into account. The patients were specifically informed about the importance (in terms of benefit) of using the iCardioMonitor (expectancy performance) and were instructed on how to adequately use the web app, including correct self-recording of the signs and symptoms (facilitating conditions). Healthcare professionals recommended regular follow-ups of their HF (social influence) [44]. Schroeder et al. [45] reported that the influence of the general practitioner upon elderly patients in relation to the use of health applications plays a crucial role, with a positive effect on patient perception of such tools. This influence may be even more significant than that exerted by other closer individuals, such as relatives, for example. Although in the present study, the role of mediators in using the iCardioMonitor was carried out by other healthcare professionals, there is some similarity in that they were the referring physicians for their chronic condition.

The results of our study showed no differences in usability scores due to educational level or age. This implies that the iCardioMonitor could be used by a large percentage of patients independently of their level of education and age, thus demonstrating adequate usability for the targeted population. However, differences were observed in terms of gender, the time since HF diagnosis, and greater self-completion of the symptoms questionnaire. The first of these results must be interpreted with caution since the number of participating females was very low, raising the possibility of bias. On the other hand, we think it makes sense that usability was rated more positively when the diagnosis of HF was closer and when patients entered more data in the symptom questionnaire. These data could be explained by the fact that greater perception of susceptibility to worsening due to less control over the disease, which is a reasonable circumstance when the diagnosis is more recent, in turn, predicts greater adherence to self-care behaviors [35]. In concordance with other authors [45], increasing perceived susceptibility to an adverse event in patients with HF contributes to increased control exerted upon the disease and, thus, to the prevention of complications.

The adequate usability scores of the iCardioMonitor web app indicate that the latter can be classified as excellent and may have positive effects on the targeted population. The transmission of knowledge and the use of devices that facilitate self-monitoring are among the main features of the iCardioMonitor and contribute to patients’ acceptance and responsibility for their condition [46]. 

As a limitation, although the results of our study demonstrate proper acceptance and usability of the iCardioMonitor, they highlight the need to assess the effectiveness of the latter in improving self-management. It thus seems necessary to develop a pragmatic trial to verify the hypothesis that the iCardioMonitor can contribute to preventing complications of HF and hospitalizations.

## 5. Conclusions

The iCardioMonitor system was developed as a web app and a knowledge-base algorithm with the capacity to predict decompensated HF. Its development has been carried out according to previously validated methods. The results obtained show that iCardioMonitor is a tool accepted by patients and has obtained a remarkable score on the usability scale. It thus may be of use in self-monitoring those symptoms consistent with decompensation of the disease. Future studies should assess the effectiveness of iCardioMonitor for early detection of complications due to HF.

## Figures and Tables

**Table 1 healthcare-12-01986-t001:** Personal characteristics and self-care score of this study’s sample (*n* = 45).

Variable	Value
Average age in years (SD)	55.8 (10.582)
Average time since diagnosis, years (SD)	7.1 (7.471)
Gender, %	
Males	88.9
Females	11.1
Educational level, %	
Primary or less	36.4
Secondary	50.0
University	13.6
Marital status, %	
Single	29.5
Married/partner	63.6
Divorced/separated	4.5
Widowed	2.3
Cohabiting, %	
Alone	13.6
Accompanied	86.4
Self-care, average (SD)	30.0 (6.654)

## Data Availability

Data underlying this article will be shared upon reasonable request to the corresponding author.

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
