# Peer review of "iCardioMonitor Digital Monitoring System for People with Heart Failure: Development and Evaluation of Its Accessibility and Usability"

_healthcare, 2024, doi:10.3390/healthcare12191986_

Round 1

Reviewer 1 Report

Comments and Suggestions for Authors

We express our gratitude to the authors for submitting their manuscript to our journal. This review focuses on the use of the iCardioMonitor System in patients with heart failure. The results are presented comprehensively, and the conclusions drawn are substantiated by the findings. However, we request the following minor revisions to enhance the manuscript:

1. To improve the introduction's quality, we recommend citing the following review: Tedeschi, Andrea et al. “Heart Failure Management through Telehealth: Expanding Care and Connecting Hearts.” Journal of Clinical Medicine vol. 13, no. 9, 2592, 28 Apr. 2024, doi:10.3390/jcm13092592. This reference provides a valuable overview of the primary telehealth methods applied in heart failure management.

2. We advise the creation of a graphical abstract summarizing the key messages of the study to facilitate readers' understanding.

3. It is essential to include a table detailing the characteristics of the 45 patients involved in the study, encompassing sex, age, BMI, cardiovascular risk factors, comorbidities, and therapy.

4. The methods section should specify whether the selected patient population includes those with HFrEF or HFpEF.

5. Finally, it is necessary to mention the specificity and sensitivity of the tested algorithm regarding its ability to detect heart failure exacerbations based on patient-reported data.

We believe these revisions will significantly enhance the clarity and impact of the manuscript.

Author Response

We express our gratitude to the authors for submitting their manuscript to our journal. This review focuses on the use of the iCardioMonitor System in patients with heart failure. The results are presented comprehensively, and the conclusions drawn are substantiated by the findings. However, we request the following minor revisions to enhance the manuscript:

  1. To improve the introduction's quality, we recommend citing the following review: Tedeschi, Andrea et al. “Heart Failure Management through Telehealth: Expanding Care and Connecting Hearts.” Journal of Clinical Medicine vol. 13, no. 9, 2592, 28 Apr. 2024, doi:10.3390/jcm13092592. This reference provides a valuable overview of the primary telehealth methods applied in heart failure management.

Thank you for the suggestion. We have included a paragraph in the introduction citing the recommended article.

  1. We advise the creation of a graphical abstract summarizing the key messages of the study to facilitate readers' understanding.

Thank you for your suggestion. We made a graphical abstract

  1. It is essential to include a table detailing the characteristics of the 45 patients involved in the study, encompassing sex, age, BMI, cardiovascular risk factors, comorbidities, and therapy.

Thanks for the suggestion. All personal information that we have is included in table 1.

  1. The methods section should specify whether the selected patient population includes those with HFrEF or HFpEF.

The selected patient population includes those with HFrEF

  1. Finally, it is necessary to mention the specificity and sensitivity of the tested algorithm regarding its ability to detect heart failure exacerbations based on patient-reported data.

Unfortunately, the size of the sample (n=45) does not allow to obtain a significant specificity or sensitivity analysis.

Reviewer 2 Report

Comments and Suggestions for Authors

HF as described is amongst the highest mortality and morbidity health care conditions throughout the world. 

The idea of cardio-monitor web based app for monitoring HF decompensation episodes is commendable but in this era of invasive monitoring (cardiomems) how reliable is an app in preventing decompensation episodes. 

The questions involved in algorithm are standard questionnaire for HF exacerbation and have the authors considered comparing the results with +/- intervention based on questionnaire used. 

Also how did the authors avoid HF admission with the questionnaire used?

Love the idea of monitoring system but what is the real life use case scenario?

What steps are taken in monitoring, treating, managing such patients?

Unfortunately it is a well thought concept but the practicality is limited. Need further explanation on this concept. 

Comments on the Quality of English Language

Minor editing of the grammar required. Nothing major

Author Response

1.The idea of cardio-monitor web-based app for monitoring HF decompensation episodes is commendable but in this era of invasive monitoring (cardiomems) how reliable is an app in preventing decompensation episodes. 

More invasive techniques were discarded in a starting point of our project to avoid patient’s distrust or discomfort. It should be considered that many of the HF patients in Spain are elderly and digital immigrants. Therefore, introducing this kind of elements, it could affect the accessibility of the app, which is one of our goals in this work.

  1. The questions involved in algorithm are standard questionnaire for HF exacerbation and have the authors considered comparing the results with +/- intervention based on questionnaire used; 3. Also how did the authors avoid HF admission with the questionnaire used?

In effect, we have preliminary results of the alerts; however, since this was not the objective of the study and the sample was insufficient to determine effectiveness, we decided not to include them in the present study. Nevertheless, we are developing a pragmatic clinical trial to determine effectiveness. Among the variables for analysis, determination is made of whether the alerts are effective in detecting cases of worsening, thus avoiding hospital admissions.

  1. Love the idea of monitoring system but what is the real-life use case scenario?

Thanks for the question. iCardioMOnitor is more than a monitoring system that, by including a predictive algorithm, is therefore based on just in time identifications of symptomatology decompensations that will allow clinicians to take immediate action upon detection, thus contributing to prevent complications of HF and hospitalizations.

  1. What steps are taken in monitoring, treating, managing such patients?

Thank you, to answer this question, we have rewritten the following sentence “The data recorded by the patients were automatically analyzed by an algorithm developed ad hoc for the iCardioMonitor and able to detect symptoms of decompensated HF (alerts). Any alert detected was automatically transmitted by e-mail to the Cardiology Department, including the patient's code, thus preserving the patient's identity (psuedoanonymization). Subsequently, the cardiology professionals, after re-identifying the patient, who in turn would contact the patient

  1. Unfortunately it is a well thought concept but the practicality is limited. Need further explanation on this concept. 

Thanks, as indicated in the questions 2 & 3, the aim of the present research was to determine effectiveness. Future research, are going to assess the effectiveness and therefore, clarify the potential of iCardioMonitor.

Reviewer 3 Report

Comments and Suggestions for Authors

The authors present a digital monitoring system for people with heart failure. The proposed methodology consists of two stages i.e., developing a monitoring system and a cross-sectional descriptive study.

The paper lacks any scientific contribution and findings. The first stage, i.e., the development of the monitoring system does not have any novel aspects in terms of algorithmic improvements or any other novel aspects. The term algorithm has been used repeatedly whereas no detail is found on whether it is a knowledge base or machine learning-based trained system. How does the system help the patients in monitoring other than collecting data?

Section 2.3.1 is very brief. The details need to be included here. Which information was collected from the literature, and how the data was validated by the experts. Parameters for inclusions and the evaluation criteria of the parameters need to be included.

The accessibility of the system is considered as a parameter (also used in the title). Three lines (196-199) are too brief to explain the parameter. How the parameter is measured, the details are missing.

The second stage i.e., the cross-sectional study measures the usability using sample data. The details of the questionnaires used are missing. The main objective of the study is to monitor the patients, however there is no discussion and results that cover this aspect of the study.

There exists a strong literature on the subject matter including automated systems using IoT and other Body Area Network devices. The authors need to include a detailed section covering those methods including comparative analysis that leads to problem statement and the motivation for the new system. In the current version, no such content is presented.

Author Response

The authors present a digital monitoring system for people with heart failure. The proposed methodology consists of two stages i.e., developing a monitoring system and a cross-sectional descriptive study.

1.The paper lacks any scientific contribution and findings. The first stage, i.e., the development of the monitoring system does not have any novel aspects in terms of algorithmic improvements or any other novel aspects. The term algorithm has been used repeatedly whereas no detail is found on whether it is a knowledge base or machine learning-based trained system. How does the system help the patients in monitoring other than collecting data?

At first time, the algorithm was designed based on the professional knowledge of an expert board supporting our research. Machine learning or more invasive techniques were discarded in a starting point of our project to avoid patient’s distrust or discomfort. Anyway, with the evolution of the accessibility of the AI all around, in a recent future we hope that the use of machine learning algorithms in medical apps would be accepted for the patients. And our system would be adapted to these techniques.

2.Section 2.3.1 is very brief. The details need to be included here. Which information was collected from the literature, and how the data was validated by the experts. Parameters for inclusions and the evaluation criteria of the parameters need to be included.

Thank you. We have rewritten the section according to the recommendation made by the reviewer.

3.The accessibility of the system is considered as a parameter (also used in the title). Three lines (196-199) are too brief to explain the parameter. How the parameter is measured, the details are missing.

Thanks for the suggestion. We have completed the paragraph. Unfortunately, there is not much more information to add as the information was provided automatically after the patients' use of the system.

4.The second stage i.e., the cross-sectional study measures the usability using sample data. The details of the questionnaires used are missing. The main objective of the study is to monitor the patients, however there is no discussion and results that cover this aspect of the study.

Thanks for the suggestions. The details of the questionnaire (SUS) was included in section 2.4. In addition, is important to notice that the aim of the present research is to develop and evaluate the accessibility and usability not to assess the effectiveness of the iCardioMonitor. Nevertheless, we are developing a pragmatic clinical trial to determine effectiveness. Among the variables for analysis, determination is made of whether the alerts are effective in detecting cases of worsening, thus avoiding hospital admissions.

5.There exists a strong literature on the subject matter including automated systems using IoT and other Body Area Network devices. The authors need to include a detailed section covering those methods including comparative analysis that leads to problem statement and the motivation for the new system. In the current version, no such content is presented.

Thank you for your comment. Indeed, there is a large literature on the existence of automated systems. We have included a reference to these systems in the manuscript. Implanted therapeutic devices are considered more complete, complex and automatic direct monitoring systems but also more expensive and less accessible to the general population. For the development of iCardioMonitor, we have opted for a home telemonitoring device that favors: 1) Equity. The cost of the iCardioMonitor is affordable for a health system of the characteristics of the system where the study was carried out, and the population is not required to wear or acquire one or more wearables; 2) Pragmatism. A system has been developed that, if proven usable and effective, can be implemented immediately; 3) Contribution to awareness. There is evidence that if the population becomes familiar with information about their disease, awareness increases and, secondly, self-care improves.

Round 2

Reviewer 3 Report

Comments and Suggestions for Authors

The authors have submitted a response document covering different points however the manuscript is not updated accordingly. Most of my concerns raised in the first version still remain un-addressed.

Author Response

(10) Reviewer last feedback was "The authors have submitted a response document covering different points however the manuscript is not updated accordingly. Most of my concerns raised in the first version still remain unaddressed".
Please try to revise following the comments of 3rd reviewer.

Thanks. Please find attached the comments and the answer to 3rd reviewer.  In addition, we include new comments (NC).

1.The paper lacks any scientific contribution and findings. The first stage, i.e., the development of the monitoring system does not have any novel aspects in terms of algorithmic improvements or any other novel aspects. The term algorithm has been used repeatedly whereas no detail is found on whether it is a knowledge base or machine learning-based trained system. How does the system help the patients in monitoring other than collecting data?

First answer: At first time, the algorithm was designed based on the professional knowledge of an expert board supporting our research. Machine learning or more invasive techniques were discarded in a starting point of our project to avoid patient’s distrust or discomfort. Anyway, with the evolution of the accessibility of the AI all around, in a recent future we hope that the use of machine learning algorithms in medical apps would be accepted for the patients. And our system would be adapted to these techniques.

NC: 1) It is a knowledge base system, and we have include it in the text. 2) How it helps patients: iCardioMonitor has the potential to detect decompensations. As literature suggests it could contribute to prevent hospitalizations and other complications. Another key point is that the system is based on the paradigm of personalized medicine because, based on the data analysed by the algorithm, clinicians could find the best solution to each patient based on their personal needs. In any case, the reviewer suggest that the system doesn’t have novel aspects. We can’t answer it because we consider that it is an opinion. Probably it could be obsolete in some parts of the world but, in our opinion, the results could contribute to improve the scientific knowledge and, therefore, to increase the evidence. i.e.: what are the preferences of Spanish population? iCardioMonitor have different characteristics that other systems that could be taken into account by developers…

2.Section 2.3.1 is very brief. The details need to be included here. Which information was collected from the literature, and how the data was validated by the experts. Parameters for inclusions and the evaluation criteria of the parameters need to be included.

First answer: Thank you. We have rewritten the section according to the recommendation made by the reviewer.

NC: nothing to add. We have answered the reviewer.

3.The accessibility of the system is considered as a parameter (also used in the title). Three lines (196-199) are too brief to explain the parameter. How the parameter is measured, the details are missing.

First answer: Thanks for the suggestion. We have completed the paragraph. Unfortunately, there is not much more information to add as the information was provided automatically after the patients' use of the system.

NC: as previous authors suggest and based in out previous expertise, the accessibility is assessed as we did. We don’t understand what the reviewer wants to read. If we had included a complex measure, logically we would have included a detailed explanation, such as, for example, usability. We consider that, when something is as obvious as the variable in question, adding futile text can only contribute to increasing the number of characters and even confuse readers.

4.The second stage i.e., the cross-sectional study measures the usability using sample data. The details of the questionnaires used are missing. The main objective of the study is to monitor the patients, however there is no discussion and results that cover this aspect of the study.

First answer: Thanks for the suggestions. The details of the questionnaire (SUS) were included in section 2.4. In addition, is important to notice that the aim of the present research is to develop and evaluate the accessibility and usability not to assess the effectiveness of the iCardioMonitor. Nevertheless, we are developing a pragmatic clinical trial to determine effectiveness. Among the variables for analysis, determination is made of whether the alerts are effective in detecting cases of worsening, thus avoiding hospital admissions.

NC: As previously indicated, the aim is not to monitor the patients, it was accessibility and usability.

5.There exists a strong literature on the subject matter including automated systems using IoT and other Body Area Network devices. The authors need to include a detailed section covering those methods including comparative analysis that leads to problem statement and the motivation for the new system. In the current version, no such content is presented.

First answer: Thank you for your comment. Indeed, there is a large literature on the existence of automated systems. We have included a reference to these systems in the manuscript. Implanted therapeutic devices are considered more complete, complex and automatic direct monitoring systems but also more expensive and less accessible to the general population. For the development of iCardioMonitor, we have opted for a home telemonitoring device that favors: 1) Equity. The cost of the iCardioMonitor is affordable for a health system of the characteristics of the system where the study was carried out, and the population is not required to wear or acquire one or more wearables; 2) Pragmatism. A system has been developed that, if proven usable and effective, can be implemented immediately; 3) Contribution to awareness. There is evidence that if the population becomes familiar with information about their disease, awareness increases and, secondly, self-care improves.

NC: We did not consider using IoT and other Body Area Network devices because of the reasons indicated. Therefore, it seems that has no sense to include a section covering those methods because it was the objective of our research.